# Smart Hand Sanitisers in the Workplace: A Survey of Attitudes towards an Internet of Things Technology

**DOI:** 10.3390/ijerph19159531

**Published:** 2022-08-03

**Authors:** Andrew D. Madden, Sophie Rutter, Catherine Stones, Wenbo Ai

**Affiliations:** 1Information School, University of Sheffield, Sheffield S1 4DP, UK; s.rutter@sheffield.ac.uk; 2School of Design, University of Leeds, Woodhouse, Leeds LS2 9JT, UK; c.m.stones@leeds.ac.uk (C.S.); wenbo.ai@network.rca.ac.uk (W.A.)

**Keywords:** hand hygiene, COVID, smart sanitiser, IoT

## Abstract

An online survey was circulated to employees from a wide range of organisations to gauge attitudes towards the idea of using smart hand sanitisers in the workplace. The sanitisers are capable of real-time monitoring and providing feedback that varies according to the hand hygiene behaviour of users. In certain circumstances, the sanitisers can monitor individuals, making it possible to identify workers whose hand hygiene falls below a certain standard. The survey was circulated between July and August 2021 during the COVID-19 pandemic. Data gathered from 314 respondents indicated support for some features of the technology, but also indicated concern about invasions of privacy and the possibility of coercion. Attitudes towards the possible implementation of the technology varied significantly according to certain characteristics of the sample, but particularly with age. Respondents above the median age were more likely to support the use of data in ways that could facilitate the promotion and enforcement of hand hygiene practices.

## 1. Introduction

The primary aim of the research described in this article was to explore reactions of workers and managers in non-clinical settings to the opportunities offered by a ‘smart’ hand sanitiser, both in the context of COVID-19 and more generally. Qualitative and quantitative data gathered from a sample of 314 respondents were used to determine whether responses to the statements in a Likert-style questionnaire (see Appendix A) are associated with specific characteristics of the respondents.

In clinical settings, the importance of hand hygiene (HH) is well understood, and many strategies have been developed to ensure compliance with organisational standards. Increasingly, such strategies rely on the use of ‘smart’ technology to monitor HH activity with a view to promoting recognised good practice [1]. However, while there has been extensive research into HH in healthcare settings, there has been comparatively little study made of attitudes and practices in office spaces in general.

This study was carried out before any COVID-19 vaccination programme had begun: the smart sanitiser that was its focus was developed in response to the recognised importance of non-pharmaceutical interventions in maintaining public health, particularly during a pandemic. The value of the findings reported here is therefore twofold. Firstly, most studies of smart HH technologies have concentrated on clinical environments. Here by contrast, consideration is given to how they might be received in other work environments. Secondly, the timing of the study makes it significant. In recent years there has been a lot of discussion about Internet of Things (IoT) technologies (such as the smart sanitiser discussed here). Much of this discussion has been about concerns that affect the acceptability of such technologies [2]. This study was carried out at a time when, because of COVID-19, ideas and innovations that could improve public health were actively encouraged. It is likely, therefore, that the findings of this study represent a high point for the acceptance of IoT technologies.

### 1.1. Non-Pharmaceutical Interventions

Much of the news linked to the spread of COVID-19 around the world has focused on medical advances that came about in response to the pandemic, particularly the development of RNA and adenovirus vaccines [2]. Alongside such pharmaceutical advances has come the development of strategies that focus on identifying and promoting appropriate behaviours.

The systematic and evidence-based deployment of vaccines (e.g., [3]) together with the implementation of programmes to overcome resistance to their use [4], has highlighted the value of initiatives that modify behaviour in ways that help to mitigate or prevent harm. Such strategies though, are far from new. Many of the measures taken to control the coronavirus outbreak are non-pharmaceutical interventions (NPIs) similar to those adopted a century earlier at the time of the Spanish flu outbreak. Such interventions form the backbone of any public health campaign, and were summarised by Galdston shortly after the Spanish Flu pandemic as cleanliness, education, and the deployment of established knowledge [5] Cleanliness, particularly in the form of HH, is now promoted in many countries by the legal obligation for employers to provide adequate handwashing facilities (e.g., [6]). It was, however, widely practiced at the time of Galdston. What has clearly changed since then has been the speed at which established knowledge can be deployed. Twenty-first-century technologies (such as smart devices) facilitate the rapid collection, dissemination, and interpretation of data, making it possible to direct behaviours in ways (such as informative messaging [7]) that were impossible during any previous pandemic.

### 1.2. Importance of Hand Hygiene

Hand hygiene is now standard practice in workplaces around the world and is recognised to be “an easy, affordable, and effective way to prevent the spread of germs and keep employees healthy” [8]. As a result, it has now become normalised to the extent that it can be difficult to determine how well guidelines are followed: self-reports of hand cleaning may represent intention rather than practice [9]. Unless people have regular reminders, they will often only clean hands that look or feel dirty [10] rather than acting in response to the invisible threat of germs [11].

In healthcare environments, managers are fully aware of this problem. Healthcare-associated infection (HCAI) is a universally recognised problem, and clear, well-researched protocols are widely available (e.g., [12]). Despite these factors, compliance among healthcare workers is often low. Erasmus et al. [13], in a systematic review, found that most reported compliance rates (defined as “the sum of all events in which hand hygiene was performed, divided by the sum of all possible hand hygiene events”) were below 50%.

Reasons for failing to comply can be summarised as ignorance (people are unaware of protocols or the reasons for them), oversight (people forget to follow protocols), and choice [14]. The latter occurs when protocols are considered detrimental (e.g., because of skin care issues), unnecessary, or impractical, or when healthcare workers are under excessive time pressure [15].

Many of these issues can be addressed by implementing measures that highlight when protocols are ignored or overlooked. Direct observation (DO) by trained auditors has been described as the “gold standard” method, but it is time-consuming, expensive, subject to variation between auditors, relies on small samples, and generates a Hawthorne effect [16]. In an online survey that took place between 2018 and 2019, 58% of healthcare workers “did not believe DO to be an effective or helpful HH measuring tool”, while 71% believed in the potential usefulness of an innovative system “that accurately recorded all HH opportunities taken, and provided the user with personal, objective and timely feedback on individual HH performance” [16].

### 1.3. Smart Technologies

Smart devices that can respond to environmental cues and message users have been around since the 1990s [17].

Given the recognised importance of hand hygiene in clinical settings, it is not surprising that hospitals were early adopters of technologies that could monitor HH activity [18]. Despite their lack of sophistication, early Smart devices proved effective in improving hand hygiene compliance [19,20,21]. Significant further developments were made possible by the growing acceptance of alcohol-based sanitising gel as an effective alternative to soap and water [22]. This made it possible for devices not only to record data relating to use, but also to dispense an effective cleaning product [18,23]. More sophisticated electronic monitoring systems have now been developed that are able to track individual employees and to alert them if they fail to sanitise their hands when required [24].

### 1.4. Hand Hygiene in Non-Clinical Settings

Smart sanitisers have been tested primarily in clinical settings, where the need for high levels of hand hygiene is generally accepted. Away from such settings, even where washing facilities are widely available, it remains a challenge to ensure that HH guidelines are followed [9,25]. As a result, comprehensive public education campaigns were deemed necessary throughout the world in order to emphasise the value of HH in mitigating the effect of COVID-19. However, even with such campaigns and with the motivation of a pandemic, adherence to recommended guidelines often falls short. Haston et al., for example, found that almost 35% of men and 23% of women surveyed during the COVID-19 outbreak admitted that they did not wash their hands after coughing, sneezing, or blowing their nose [26]. Furthermore, the association between self-reported and observed hand hygiene adherence has often been found to be low, e.g., [27], so despite the intense focus on hand hygiene at the time, it is likely that this is an underestimate.

In clinical settings, high standards of hand hygiene are crucial at all times. During a pandemic, they become crucial in every public space, especially work spaces where there is prolonged and frequent contact between colleagues. However, even before the COVID-19 outbreak, health benefits of a more proactive approach to HH in office environments were apparent [28], and advantages may extend beyond health. Zivich et al., for example, identified both financial and social benefits [29]. The latter may have been increased by COVID, which has opened up areas of possible conflict. In a survey of workers carried out by the Resolution Foundation in September 2020, 47% of black, Asian, and minority ethnic workers reported an active concern about the transmission of COVID-19 in the workplace compared to 34% of white employees. This is one of a number of discrepancies in attitudes [30] that could be a source of friction amongst colleagues if some are thought not to be cleaning their hands adequately.

There are, therefore, potential advantages to the kind of behavioural modification that might be facilitated by smart sanitisers. If the sanitisers are able to share their data on the Internet of Things (IoT), further possibilities emerge.

### 1.5. IoT

The internet is generally taken to have begun in 1969 [31]. Within thirty years, it had radically changed the way that people shared information, with profound effects on commerce, government, and education across the developed world. In 1999, another radical change occurred when Proctor & Gamble started using Radio Frequency IDentification (RFID) in their supply chain. Kevin Ashton, a brand manager at the company, gave a presentation in which he predicted the advent of an Internet of Things (IoT), where devices collected a variety of data and shared it on the internet [32]. One significant impact of such a development would be a fundamental change in the nature of the internet. As devices became smarter and able to gather more and more data about their use and their environment, information flows between online devices would eventually exceed information flows between people.

### 1.6. IoT Threats and Opportunities

Commentators were not slow to warn of threats to privacy posed by the IoT. Numerous privacy concerns have been raised [33], and these are exacerbated by concerns about security. Cyber security expert Mikko Hyppönen argues that “if an IoT device is ‘smart’, it’s vulnerable”, and has described internet-connected smart devices as the asbestos of the future: “a great innovation, which then decades later turned out to be the worst innovation” [34].

However, it is the opportunities afforded by online smart devices that have generated most discussion. In recent times, much of that discussion has centred on the potential of IoT technologies to combat the spread of COVID-19 (e.g., [35]). The smart sanitiser that is the focus of this study was developed with this in mind, but as discussed above, the promotion of good HH has an important public health role in general, and the ability to monitor HH behaviour makes several courses of action possible. Smart hand sanitisers collect data which organisations can use to determine when and where employees wash their hands; where best to place sanitisers; and when to replenish or repair them [36,37]. Some smart devices are also capable of displaying messages that are modified according to the behaviour of individual users [7]. Others, rather than displaying messages, display video evidence to indicate how thoroughly hands are cleaned [38]. Opportunities increase still further if the device contains an RFID reader. Organisations whose members are required to carry RFID identity cards are then able to monitor aspects of the HH behaviour of individuals and to identify employees whose behaviour falls below a required standard [39].

## 2. Methods

The research reported here resulted from a collaboration between the University of Sheffield (Information School) and the University of Leeds (School of Design), together with Savortex^®^, a manufacturer of IoT hand hygiene technology, including hand sanitisers.

Data collection began in July 2021 during the COVID-19 pandemic. As a result, researchers were limited in their movements, so all data were collected online. Potential volunteers were given details of the project and information about its objectives, and were only able to participate if they ticked a box to indicate that they had been informed about the project and that their participation was voluntary.

Surveys were used to obtain both quantitative and qualitative data, and additional qualitative data were collected through online interviews. An abductive approach was adopted, in which data gathered at each stage were used to shape subsequent data collection instruments. Abduction, developed from the work of Charles Peirce [40] (who also referred to it as ‘hypothesis’), considers findings that appear exceptional and infers a set of circumstances in which the findings would be normal. This makes such an approach ideal for exploratory mixed methods studies, where statistically significant findings can be contextualised using qualitative data.

The primary data collection tool was an online survey (Appendix A). Themes for the survey were derived from information notices provided for the public [6,41] and from research literature. A pilot study of the survey gathered comments and suggestions from fifteen volunteers. Their responses were not included in the final dataset, but they did lead to significant improvements in the survey. The revised version was distributed using a combination of convenience sampling and snowball sampling. Though not as generalisable as studies based on probability sampling strategies [42], these methods of data collection were more practical (particularly given the constraints imposed by the pandemic), and are useful for the kind of exploratory study being presented here [43]. The survey was circulated to names on Savortex^®^’s customer database and to the University of Sheffield’s volunteer database. In addition, it was sent to professional and personal associates of the authors.

The questions in the survey were almost entirely quantitative and none were obligatory. Most were Likert-style questions with a 7-point scale. This scale was selected in accordance with the findings of Preston and Colman [44], which suggest that “rating scales with 7, 9, or 10 response categories are generally to be preferred.” When selecting significance, a strict level of *p* < 0.01 was used. Given the high number of variables being assessed, the standard measure of *p* < 0.05 was considered too likely to result in Type I errors [45].

The last part of the survey form, however, asked respondents to provide an email address if they were willing to help further with the research, and invited them to enter observations or suggestions regarding the topics covered by the questionnaire. This elicited 52 usable responses, which were thematically analysed, with codes being constructed inductively [46]. Findings were used to inform a follow-up qualitative questionnaire (Appendix B) and an interview script. People who had left their email address were contacted and asked either to complete the additional questionnaire or if they were willing to be interviewed. Twelve qualitative questionnaires were returned, and three people were interviewed. The data provided by Appendix A (21), Appendix B, and by the three interviews, gave qualitative details that help to contextualise findings from the quantitative survey.

One negatively coded statement was included in the questionnaire, together with two statements that were conditional: i.e., people should only respond if they were in a managerial position. Prior to analysis, the dataset was cleaned [47]. Responses were considered unusable if

a respondent strongly agreed with over 70% of statements (including the negatively coded statement);respondents who stated that they had a non-managerial position strongly agreed with over 50% of statements (including the negatively coded statement), and responded to the conditional statements.

These were taken to indicate that the survey statements were not being properly evaluated; and on the basis of these conditions, 13 sets of responses to survey Appendix A were classed as unengaged responses and eliminated from the analysis, leaving 314 sets of responses to be analysed.

## 3. Results

An exploratory analysis of the data is presented below. This summarises the sample profile and seeks to determine whether responses to the statements in Appendix A (12–19) are associated with particular characteristics of the respondents. No item in either survey was compulsory, so where a respondent chose not to provide information, the value of n was less than 314.

### 3.1. Sample Characteristics

Respondents were predominantly white (87%), UK-based (88%), female (71%), and ranged in age from 21 to 72 (Table 1).

Most respondents (53%) were in non-managerial roles (Table 2), and most were working in the public sector or in education (64%) (Table 3).

Around a quarter (23.6%) had some responsibility for hand hygiene equipment (either installation or maintenance), or for health and wellbeing within their organisation. The majority of respondents (63.5%) were employed in workplaces with a hundred or more staff (Table 4), and most (66.9%) estimated that workplace attendance dropped to 25% or less at the peak of the pandemic.

A minority predicted a full return to work after the vaccination programme, but most respondents (57.6%) estimated that over 50% of employees would return (Table 5). At the time of responding (between July and August 2021), little more than a quarter of respondents (27.5%) were attending the workplace (Table 6).

Table 7 is an attempt to capture respondents’ views on the extent and direction of change in workplace attendance during the pandemic. Responses were coded from 0 to 4, where 0 indicates an estimate of 0% attendance, and 4 indicates an estimate of 76–100% attendance. Codes assigned to estimated attendance during lockdown were subtracted from the codes assigned to estimated attendance after vaccination. The resulting integer is taken to be an indication of how much the respondent feels attendance would change.

### 3.2. Issues Identified

The qualitative question at the end of the survey (Q21—see Appendix A) gave respondents the opportunity to identify factors that were of interest or of concern to them. Fifty-two people gave non-trivial responses to Q21. A thematic analysis was carried out on these, and key themes were used to create a qualitative questionnaire (Appendix B) and an interview script. Themes identified from these three data-collecting exercises are listed in Table 8, along with indicative quotations. These were of value in interpreting the responses to statements summarised in Figure 1, Figure 2, Figure 3, Figure 4, Figure 5, Figure 6 and Figure 7.

### 3.3. Responses to the Statements

Responses to the Likert-style statements are summarised in Figure 1, Figure 2, Figure 3, Figure 4, Figure 5, Figure 6 and Figure 7 below. To simplify the charts, some responses have been grouped together. For example, in Figure 1, people were asked for their response to a statement on a scale from 1 (strongly disagree) to 7 (strongly agree). To generate the chart, scores of 1–3 were taken to indicate disagreement, scores of 5–7 were taken to indicate agreement, and a score of 4 was treated as neutral. The same approach was used in preparing Figure 2, Figure 3, Figure 4, Figure 5, Figure 6 and Figure 7.

Figure 1, Figure 2, Figure 3 and Figure 4 suggest general acceptance of the importance of hand hygiene. For most of the statements, over 80% of respondents shared similar views, agreeing with positive statements, and disagreeing with the negative assertion that high standards of hand hygiene were not relevant to their organisation (Figure 1). Areas where there was less consensus can be accounted for by some of the themes identified in the interviews and in response to the questions in Appendix B (summarised in Table 8). Most prominent amongst these are

Sensitive skinSanitisers in a broader hand hygiene contextCoercionConcerns about monitoring

Concerns about sensitive skin and about the need to consider hand sanitisation within a broader context almost certainly influenced responses to the statement “Well-established hand washing facilities, such as soap and water, are good enough”. The fact that a small majority (56%) were in agreement (Figure 2) suggests that there may be reservations about any hand hygiene strategy based solely on sanitisers.

Respondents’ concerns about coercion presumably account for the fact that only a minority (42%) of the sample agreed that “If colleagues repeatedly fail to clean their hands properly, management should be able to take action to prevent them coming into contact with the rest of the workforce” (Figure 4). Similar concerns may also contribute to the large minorities that were not in agreement with statements such as “An organisation should ensure that its employees maintain a minimum standard of hand hygiene” (Figure 3) and “As a result of COVID-19, my employer should be prepared to introduce measures that ensure compliance with handwashing guidelines” (Figure 4).

The fear that monitoring would lead to an invasion of privacy probably influenced the 48% of respondents who did not agree that “It is important for employers to have an accurate picture of how well their employees clean their hands” (Figure 3). It almost certainly impacted statements relating to monitoring (Figure 6 and Figure 7). While there was broad acceptance of the idea that data collected by smart sanitisers could “provide maintenance staff with up-to-date information about whether or not sanitisers are working properly” (Figure 6), the use of data to target individuals (even if only to tailor messages) was less popular. However, it should be noted that although opinions clearly varied, responses indicated that there was support for measures that may result in a loss of privacy, and could be considered coercive.

Figure 5 was interesting because it indicated that levels of concern about transmission in the workplace were higher when managers were expressing views about the workforce than when the workforce was expressing views about itself. The last two statements in Figure 5 were targeted specifically at managers. As discussed below, the managerial level was not significantly related to responses to any statements, and so was presumably not a factor in responses to the first four statements.

### 3.4. Significant Associations between Sample Characteristics and Response to Statements

#### 3.4.1. Age

The quantitative questionnaire included 36 Likert statements targeted at everyone (plus two statements targeted only at managers). For 18 of the 36 statements, there were statistically significant correlations between the ways in which people of different ages responded (Table 9 and Table 10).

These give an indication of the extent to which the views of respondents above and below the median age of 46.5 differed. Some of the difference is because younger people were more likely to respond with a neutral score, but in many cases, there were marked differences between levels of agreement. These are shown in Table 11, where the statements are ranked according to the difference between the percentages of younger and older respondents agreeing with each one.

A review of Table 11 indicates that the statements where older (above median age) and younger (below median age) respondents were most likely to disagree related to the promotion and enforcement of hand hygiene practices, and the use of data in ways that might facilitate these. In most cases, where the majority of older respondents agreed with a statement, so too did younger respondents. The same is also true where the majority of respondents disagreed with a statement. In a few cases, though, majority opinions differ. The majority of older respondents agreed with the following statements, while most younger respondents disagreed or gave a neutral response (Table 10):*It is important for employers to have an accurate picture of how well their employees clean their hands.**If colleagues repeatedly fail to clean hands properly, management should be able to take action to prevent them coming into contact with the rest of the workforce.**Monitoring of hand hygiene at work... would signal management’s commitment to employee wellbeing.*

By contrast, where most younger respondents were unhappy with the following potential uses of smart sanitiser data, older respondents were either happy or neutral about such uses.


*Data collected by Smart hand sanitisers can... display personalised messages to a user, based on that user’s previous hand hygiene behaviour.*

*Data collected by Smart hand sanitisers can... help to identify individuals whose hand cleaning behaviour is below an acceptable level.*


#### 3.4.2. Sex

Men and women differed little in their responses to the various statements. Fisher’s exact test was used in place of the χ^2^ test because the expected cell count was below 5 in many cases [48]. Responses to two statements were significantly related to the sex of the respondent (*p* < 0.01).


*Promoting good hand hygiene practice is as important as providing good hand hygiene resources.*

*Levels of hand hygiene practised during the COVID-19 crisis should be maintained after the current programme of vaccination is completed.*


Although similar majorities of men and women (>80%) agreed with both the statements, women were more likely to give a score of 7, suggesting that they agreed more strongly.

#### 3.4.3. Ethnicity and Home Country

Responses were received from Asians, black people, and people of mixed ethnicity. Respondents were based in Australia, South Asia, North America, and mainland Europe. However, the overwhelming majority (82%) were white and UK-based. The proportion of non-white and non-UK respondents was therefore too low to draw any meaningful conclusions about ethnicity and home country.

#### 3.4.4. Industry Sector

As was noted above, the study sample was dominated by people employed in education and public service (64%). To determine whether their responses were atypical, the sample was divided into two parts, comprising (1) education and public service and (2) other sectors. Fisher’s exact test found no significant differences.

#### 3.4.5. Responsibility for Hand Hygiene Equipment

Just under a quarter of people responding had responsibility for one or more of the following:

Installation of hand hygiene equipment.Maintenance of hand hygiene equipment.Health/wellbeing within your organisation.

These respondents were grouped together and compared to respondents with none of these responsibilities using Fisher’s exact test. The two groups differed significantly in response to only one statement:*Strict hand hygiene regulations can disrupt work routines.*

Although most respondents disagreed with this statement, people with none of the above responsibilities were more likely to do so (82.2%, compared to 69.3%).

#### 3.4.6. Managerial Level

Level of management was treated as an ordinal variable, ranging from 1 (non-managerial) to 4 (directorial). This allowed it to be correlated against responses to the various statements. No significant Spearman correlations were detected, suggesting that managers and non-managers had similar views regarding the hand hygiene issues raised by the Likert statements in the questionnaire.

#### 3.4.7. Number of People Employed in Workplace before the Pandemic

Size of organisation had little impact on attitudes. Spearman rank correlations found only two statements where responses varied significantly with the size of organisation:*Good hand hygiene is important if my organisation is to function properly.**My employer should be interested in developments that may improve hand hygiene.*

In both cases, few people disagree with the statements, but there was a slight tendency for respondents from larger organisations to be more strongly in agreement than respondents from smaller organisations.

#### 3.4.8. Employment during COVID-19

Predicted change in attendance between peak lockdown and after the vaccination programme (Table 7) was significantly correlated (Spearman) to just one variable:*Monitoring of hand hygiene at work would signal management’s commitment to employee wellbeing.*

The correlation was negative. Respondents who predicted little or no change between attendance at peak lockdown and attendance after the vaccination programme were more likely to agree with the statement.

Further insight into whether or not attendance during lockdown impacted responses to the Likert-style statements was gained by analysing responses to the question:


*“When COVID-19 restrictions were at their strictest, which of the following was true?”*


Initially, this was treated as an ordinal variable, using the values 0 to 4 (as shown in Table 8). No significant Spearman correlations were found. The responses were subsequently reclassified by grouping 0 to 2 into Did Not Attend Workplace (0), and 3 to 4 into Attended Workplace (1). This binary classification made it possible to apply Fisher’s exact test, which found significant differences in responses to:*It is important for employers to have an accurate picture of how well their employees clean their hands.*

Levels of agreement with the statement were much the same whether respondents were or were not going into work (55% and 51%, respectively), but people attending the workplace were more likely to strongly agree (25% compared to 12%).

## 4. Discussion

### 4.1. Risk Factors

Research to date identifies the main risk factors for COVID-19 as being sex and age [49,50,51]. Ethnicity has also been identified as a risk factor for COVID-19, but appears to be mediated in part by community-level socioeconomic factors [52]. In this study, the lack of ethnic diversity among respondents makes it difficult to draw conclusions, an issue that has also arisen in other studies [51].

Given the importance of sex as a risk factor, it is slightly surprising that the responses of men and women were not more different. The two statements where there was a significant difference reflect the findings of other studies, which indicate that women tend to be more diligent in matters of hand hygiene than men [53]. However, in this study, possibly the firmest evidence for such a difference is in the response rate of men and women. The relatively low proportion of men responding (29%) supports the suggestion that hand hygiene is seen as a lower priority amongst men.

The factor that had the most impact on whether or not respondents agreed or disagreed with statements was age. Table 11 implies greater acceptance among older respondents of measures that could lead to a more prescriptive hand hygiene culture within employing organisations, suggesting that older respondents were more cautious than younger ones. This appears to contradict the findings of Judge and Slaughter [30], who found that respondents aged 55–65 were substantially less likely to report active concern about the transmission of COVID-19 in the workplace than respondents aged 18–24. However, given that the COVID-19 death rate doubles for every five to six additional years of age [50], greater caution among older people is unsurprising.

Although not usually listed as a risk factor, the size of employing organisations could contribute to the risk of catching the virus because employees of large organisations have more opportunities to encounter COVID-19 carriers. It is slightly surprising, therefore, that there was not more evidence of support for the stricter enforcement of guidelines among respondents from large organisations.

Another risk factor that is not often identified as such is culture. Obviously, it has been much discussed in association with vaccination uptake, e.g., [54], but some of the qualitative data in this study suggested that cultural factors associated with organisational, social, religious, and political differences may impact HH practices.

### 4.2. Monitoring and Behavioural Modification

The significant differences discussed above are interesting and informative. However, it is also interesting that while most proposed uses of data from the smart sanitisers were supported by a majority of respondents, the only statement for which there was little (<10%) opposition was that the data could “…*provide maintenance staff with up-to-date information*…” (Figure 6).

The qualitative insights from interviews and responses to Appendix B provided some insight into the reasons for opposition. One of the themes to emerge from the qualitative responses was the need to consider how data will be used before they are collected, and to ensure the “*need to be actioning something on the back of it*” (Table 8). In other words, data should be gathered with a view to addressing defined needs, rather than simply because the technology makes data collection possible. Several voices in the study expressed concerns about invasions of privacy and about the quality of data. A sensible way to engage with such concerns is to explain which decisions will be informed by the data and demonstrate how they will be influenced. Providing maintenance staff with up-to-date information is a clear and uncontroversial example of an area where decision making could benefit.

Statements relating to the use of data to identify individuals (Figure 6 and Figure 7) were amongst those with the lowest level of approval. Worries about privacy probably contributed to these responses, but so too did concern about the quality of the data. If smart sanitisers are to modify hand hygiene behaviour in beneficial ways, then the data need to reflect desired behaviours that are appropriate to the circumstances in which the devices will be used. For example, healthcare workers have, on occasion, failed to comply with HH guidelines because the locations of hand cleaning facilities were not convenient for the workflow processes [14].

Any assumptions made when interpreting data should be made explicit [55] so that they can be properly assessed. Many suggestions for ways in which smart sanitisers can help to identify people whose HH is inadequate are predicated on the assumption that sanitisers are the only means by which hands can be cleaned. The use of private supplies of hand cleaning gel, or of soap and water, are ignored; so too is the qualitative issue of how well hands are cleaned.

### 4.3. Limitations of the Sample

The sampling approach made it possible to reach a large number of respondents within the limitations of time and budget, but it was skewed towards education and the public services. However, the working environment of these sectors is largely representative of the office-based population, so there is still much of value that can be learned.

## 5. Conclusions

It should be remembered that the survey was carried out during a pandemic, so even in the most relaxed of working environments, significant efforts were being made to raise the standards of HH. Smart IoT hand sanitisers offer a potential means of achieving this. Under other circumstances, there may have been less willingness to accept measures that many respondents regarded as intrusive. However, bearing in mind this caveat, the analysis presented here raises the following key points:

### 5.1. Data and Decisions

There was a recognition by most respondents that smart IoT sanitisers had the potential to increase standards of hand hygiene. However, there were clear concerns about privacy. There were also concerns about the quality of data, and the extent to which monitoring data reflected HH practices.Management should not assume that sanitisers are the only means by which hands can be cleaned, since many people will wash with soap and water, or bring their own sanitising gel.Given such concerns, data should be gathered from Smart devices with a view to addressing defined needs, rather than simply because the technology makes data collection possible.If smart devices are deployed, then management should explain which decisions will be informed by the data they collect, and should demonstrate how those decisions will be influenced.

### 5.2. Perceptions of Risk

Perceptions of risk may have contributed to older respondents’ apparent willingness to accept measures that could lead to a more prescriptive hand hygiene culture.The sociocultural perspectives of some respondents may have influenced how they balanced perceived risks, with the health benefits of rigorous hand hygiene practices being set against concerns about intrusive management.

There is always a balance to be found between the possible disruption of strict HH routines and the need to prevent the spread of disease. In some workplaces (such as healthcare environments and food preparation areas), the balance is always in favour of high HH standards. In other workplaces, the balance is more variable; but smart sanitisers, with the ability to monitor unobtrusively and to vary messages in response to the behaviours of workers, have the potential to make a significant contribution to any non-pharmaceutical interventions, should future circumstances shift the balance once more.

## Figures and Tables

**Figure 1 ijerph-19-09531-f001:**
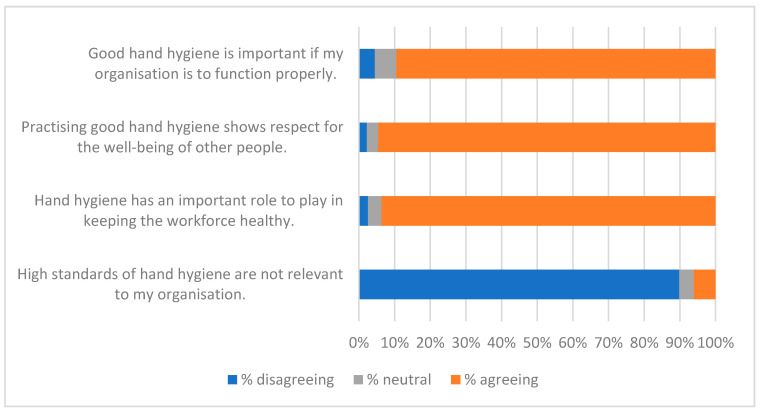
Importance of HH.

**Figure 2 ijerph-19-09531-f002:**
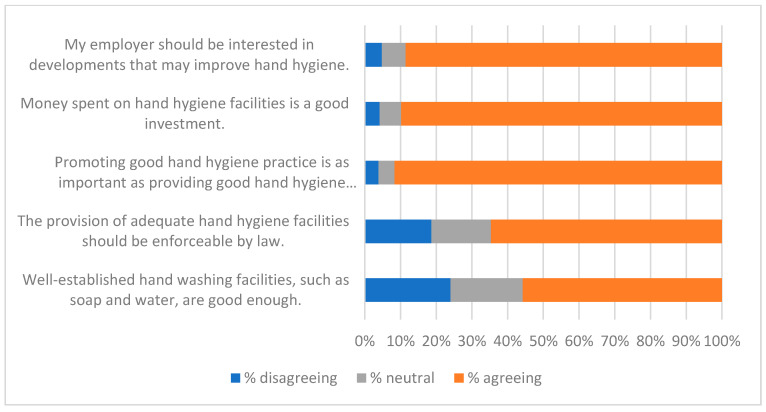
Investing in HH.

**Figure 3 ijerph-19-09531-f003:**
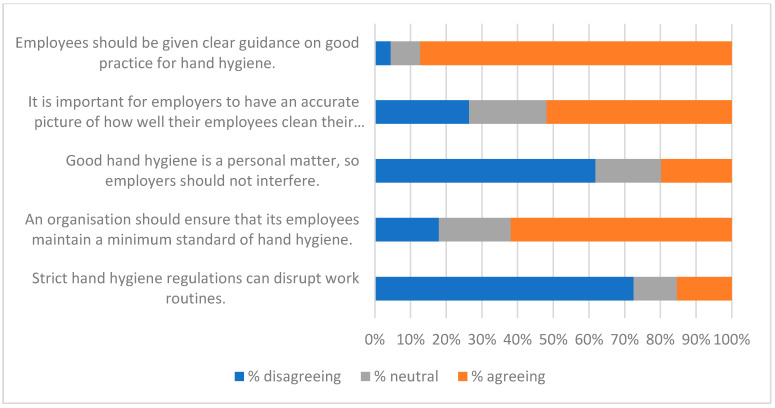
Promoting HH.

**Figure 4 ijerph-19-09531-f004:**
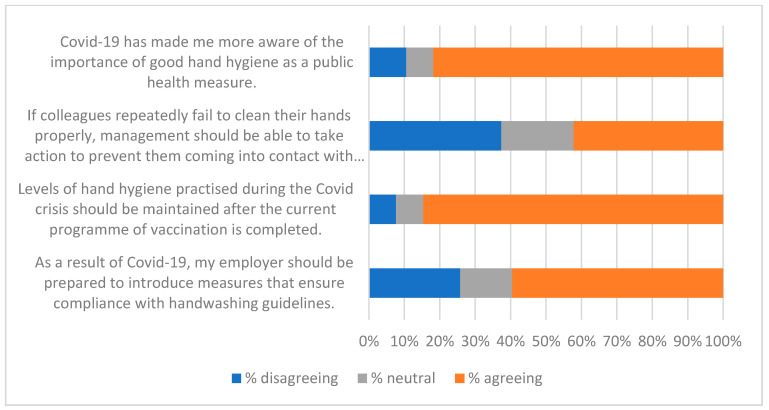
Impact of COVID-19.

**Figure 5 ijerph-19-09531-f005:**
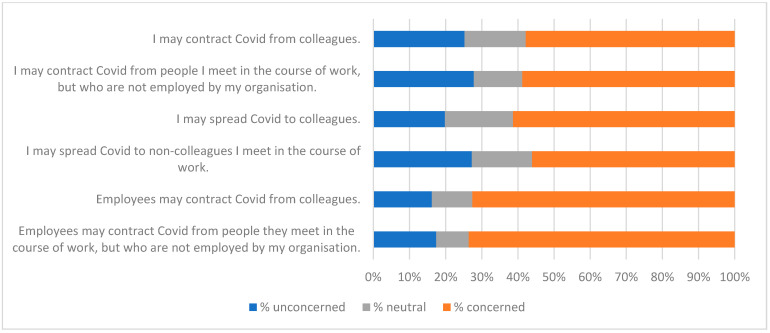
Spread of COVID-19.

**Figure 6 ijerph-19-09531-f006:**
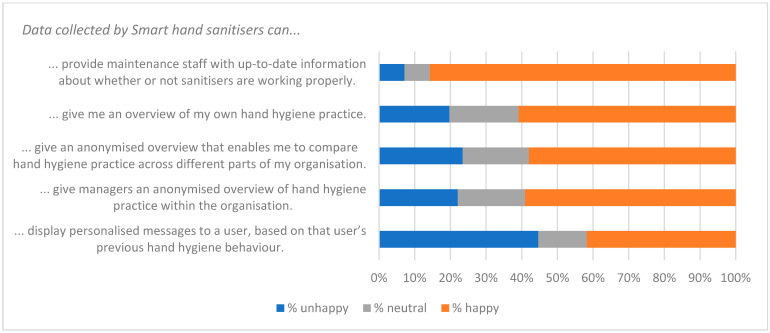
Acceptance of monitoring.

**Figure 7 ijerph-19-09531-f007:**
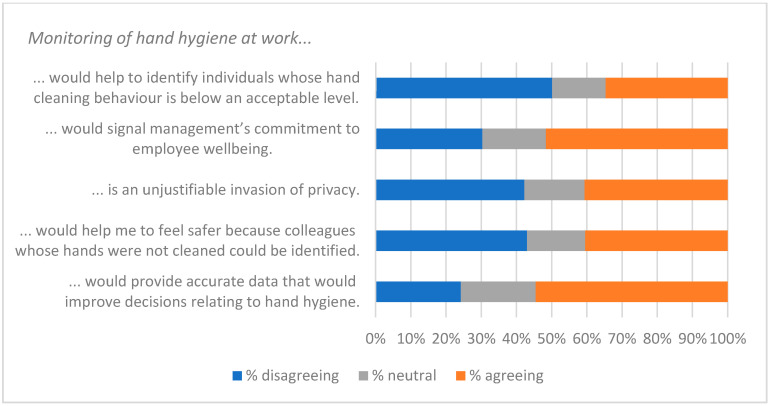
Agreement with monitoring.

**Table 1 ijerph-19-09531-t001:** Age of respondents (mean = 46.3. median = 46.5, s.d. = 11.0, n = 306).

Age	21–25	26–30	31–35	36–40	41–45	46–50	51–55	56–60	61–65	66+
%	2.6	7.2	8.2	14.1	16.0	11.4	17.6	13.4	7.5	2.0

**Table 2 ijerph-19-09531-t002:** Management level of respondents (n = 311).

Management Level	Non-Managerial	Middle Management	Senior Management	Directorial
%	53.1	33.4	9.0	4.5

**Table 3 ijerph-19-09531-t003:** Industry sector (n = 314).

Industry Sector	%
Charity/voluntary	10.2
Education	43.3
Financial and professional services	3.5
Health services	5.4
Heritage	2.2
Media, culture, graphical	4.1
Public service	20.7
Transport	2.5
Other	8

**Table 4 ijerph-19-09531-t004:** Number of employees at respondent’s organisation (n = 299).

Size of Workforce	<10	10–49	50–99	100–249	500+	Don’t Know
%	8.9	19.7	5.1	18.5	43.0	4.8

**Table 5 ijerph-19-09531-t005:** Estimated number of employees working during the pandemic (n = 280).

	0%	1–25%	26–50%	51–75%	76–100%	Can’t Say
Estimated attendance at peak lockdown (% responses)	16.6	50.3	6.1	4.5	12.1	10.5
Estimated attendance after vaccination programme (% responses)	0.6	12.1	16.2	19.4	38.2	13.4

**Table 6 ijerph-19-09531-t006:** Number of employees at respondent’s organisation (n = 314).

Status	Unemployed	Furloughed	Worked Remotely	Chose to Attend Work	Required to Attend Work
%	2.2	6.1	64.2	12.5	15.0

**Table 7 ijerph-19-09531-t007:** Predicted change between estimated attendance at peak lockdown and estimated attendance after programme vaccination has been completed (n = 262).

<0	0	1	2	3	4	Can’t Say
5.7	16.9	19.7	17.5	20.4	5.4	20.1

**Table 8 ijerph-19-09531-t008:** Themes identified from qualitative responses.

Behavioural change	“…*historical data in the UK shows that improving public awareness rather than legally enforcing some behaviours, is the long term route to improving take up.*” “*Good/best practice needs to come from the top down and if senior managers aren’t seen to be doing something those at the lower levels won’t do it either.*”
Circumstances where monitoring is acceptable	“*I think it’s a good idea to be able to monitor use and performance of hand sanitising stations, but not to be able to identify individuals.*” *“Certainly it would help our organisation stay on top of re-filling the devices, as sometimes our current devices run out of sanitiser. It could also help demonstrate when staff are and aren’t cleaning their hands at appropriate moments, for example when entering the cafeteria or when entering the building.*”
Coercion	“*…we are managed by a group of people who believe in divide and rule and who manage people through criticism and humiliation. To give these people a data set which they will simply use as a stick to beat different work units in the organisation is very foolish*.” “*…collecting data that leads to behavioral impositions would not be welcomed*”
Cultural perspective	“*Many parts of the USA have a more libertarian approach to everything, but especially things considered very personal and private such as hygiene and surveillance. The consideration in such culture is that courtesy and respect should not be coerced or monitored. Personal responsibility does not extend to interfering in someone else’s life.*” “*My answers are coloured by the fact that I work in a hospital so hand hygiene is already of critical importance.*”
Data quality	“*Hand sanitisers provide fundamentally unsafe data... by exclusively following sanitiser data, rather than sink data, you’re not really helping change public health education*.” “*People may choose to use their own hand sanitisers, or to wash their hands with soap and water instead, so the smart sanitisers may not collect an accurate picture of who is sanitising their hands and when.*”
Data use	“*…if you’re going to be collecting it [data] then you need to be actioning something on the back of it.”* *“Why would I collect that data? It’s difficult for me to visualise sitting in front of that data and doing anything with it.”*
Impact of COVID-19	“[COVID] *has given hand hygiene a much, much higher profile than it had before in the general organisation. In the laboratories there was already a very strict insistence on hand hygiene. However in most areas—offices, stores, teaching areas—there was very limited emphasis on it…*” “*Sanitiser was never previously provided, however hand hygiene was always important as extensive hand washing facilities have always been provided*”
Personal preference	“*Sanitising liquid is horrible…*” “…*I don’t like hand sanitisers and after using them I prefer to wash my hands as soon as possible.*”
Privacy concerns	“…*singling out individuals makes me feel uncomfortable and I feel is an invasion of privacy and in a workplace there should be two-way trust and respect, and feeling spied upon by your employer does not play any part in that*.” “*It’s a difficult balance between big brother and the welfare of colleagues and customers and… I do not think it would be an easy sell to my teams”*
Sanitisers in broader HH context	“*COVID-19 is a respiratory virus and the biggest risks are from not covering the nose and mouth, or close contact with potentially infected people. Focusing on only hand hygiene is missing a major contributor to the spread of this respiratory virus.*” “*The survey appears slightly biased towards ‘sanitisation’, which suggests the use of a sanitiser, whereas guidance suggests that soap and water is better when available.*”
Sensitive skin	“*The importance of hand hygiene in disease spread needs to be balanced against the wellbeing of employees with skin conditions such as eczema which may be exacerbated*” “*Hand washing is better than sanitising, which has given me dermatitis in the past.*”
Unintended consequences	“*...making it solely the responsibility of the individual by ‘invading privacy’ will create a back lash and is an easy target for misinformation on social media.*” “*I think it could easily be used as a stick rather than a carrot by poor managers so I could see that being more toxic than passing on germs in some workplaces!*”
Workplace context	“*In the course of their work some people remove our refuse and clean our public buildings, some handle cash, some operate machinery and some build our homes and infrastructure. It would be wrong to monitor people all the time and expect them to have hands as clean as a surgeon.*” “[Company name] *manufactures food and pharmaceutical labels/packaging… Comparing a conventional print manufacturing site without hygiene standards to one with our level could skew any results were that not taken into consideration.*”

**Table 9 ijerph-19-09531-t009:** Statements positively correlated with the age of the respondent. Spearman rank correlation, *p* < 0.01. (Older ⇒ > median, Younger ⇒ < median). N indicates the no. of people from the Younger or Older age group who responded to the statement.

		% Dis-Agree	% Neutral	% Agree	N
*Good hand hygiene is important if my organisation is to function properly.*	Younger	6.5	7.8	85.6	153
	Older	2.6	4.6	92.8	153
*My employer should be interested in developments that may improve hand hygiene.*	Younger	5.2	9.8	85.0	153
	Older	4.6	3.3	92.2	153
*Money spent on hand hygiene facilities is a good investment*	Younger	4.6	7.8	87.6	153
	Older	3.9	3.9	92.2	153
*Promoting good hand hygiene practice is as important as providing good hand hygiene resources*	Younger	5.3	6.6	88.2	152
	Older	2.6	2.6	94.8	153
*Employees should be given clear guidance on good practice for hand hygiene*	Younger	5.2	10.5	84.3	153
	Older	3.9	6.5	89.5	153
*It is important for employers to have an accurate picture of how well their employees clean their hands*	Younger	32.7	22.9	44.4	153
	Older	19.6	20.9	59.5	153
*An organisation should ensure that its employees maintain a minimum standard of hand hygiene*	Younger	22.2	23.5	54.2	153
	Older	13.9	15.9	70.2	151
*If colleagues repeatedly fail to clean hands properly, management should be able to take action to prevent them coming into contact with the rest of the workforce*	Younger	42.2	23.4	34.4	153
	Older	31.6	17.1	51.3	152
*As a result of COVID-19, my employer should be prepared to introduce measures that ensure compliance with handwashing guidelines*	Younger	29.4	20.3	50.3	153
	Older	21.6	9.2	69.3	153
*... give me an overview of my own hand hygiene practice*	Younger	22.4	22.4	55.3	152
	Older	17.4	14.8	67.8	149
*... give an anonymised overview that enables me to compare hand hygiene practice across different parts of my organisation*	Younger	27.8	21.2	51.0	151
	Older	17.6	16.2	66.2	148
*... give managers an anonymised overview of hand hygiene practice within the organisation*	Younger	25.7	21.1	53.3	152
	Older	16.8	16.8	66.4	149
*... display personalised messages to a user, based on that user’s previous hand hygiene behaviour*	Younger	50.7	11.2	38.2	152
	Older	36.9	16.1	47.0	149
*... help to identify individuals whose hand cleaning behaviour is below an acceptable level*	Younger	57.2	13.8	28.9	152
	Older	40.9	17.4	41.6	149
*... would signal management’s commitment to employee wellbeing*	Younger	34.6	21.6	43.8	153
	Older	24.8	14.1	61.1	149
*... would help me to feel safer because colleagues whose hands were not cleaned could be identified*	Younger	48.7	17.1	34.2	152
	Older	35.6	16.1	48.3	149

**Table 10 ijerph-19-09531-t010:** Statements negatively correlated with age of respondent. Spearman rank correlation, *p* < 0.01. (Older ⇒ > median, Younger ⇒ < median).

*Good hand hygiene is a personal matter, so employers should not interfere*.	Younger	59.5	22.2	18.3	153
	Older	62.7	15.7	21.6	153
*Strict hand hygiene regulations can disrupt work routines.]*	Younger	69.1	11.2	19.7	152
	Older	76.5	11.8	11.8	153

**Table 11 ijerph-19-09531-t011:** Statements where response is significantly correlated to age. Statements are ranked according to difference in levels of agreement between older and younger respondents.

Corresponding Figure			% Difference
Figure 4	1	*As a result of COVID-19, my employer should be prepared to introduce measures that ensure compliance with handwashing guidelines*	19.0
Figure 7	2	*... would signal management’s commitment to employee wellbeing*	17.3
Figure 4	3	*If colleagues repeatedly fail to clean their hands properly, management should be able to take action to prevent them coming into contact with the rest of the workforce*	16.9
Figure 3	4	*An organisation should ensure that its employees maintain a minimum standard of hand hygiene*	16.0
Figure 6	5	*... give an anonymised overview that enables me to compare hand hygiene practice across different parts of my organisation*	15.2
Figure 3	6	*It is important for employers to have an accurate picture of how well their employees clean their hands*	15.0
Figure 7	7	*... would help me to feel safer because colleagues whose hands were not cleaned could be identified*	14.1
Figure 6	8	*... give managers an anonymised overview of hand hygiene practice within the organisation*	13.2
Figure 7	9	*... help to identify individuals whose hand cleaning behaviour is below an acceptable level*	12.7
Figure 6	10	*... give me an overview of my own hand hygiene practice*	12.5
Figure 6	11	*... display personalised messages to a user, based on that user’s previous hand hygiene behaviour*	8.8
Figure 3	12	*Strict hand hygiene regulations can disrupt work routines.*	−8.0
Figure 1	13	*Good hand hygiene is important if my organisation is to function properly.*	7.2
Figure 2	14	*My employer should be interested in developments that may improve hand hygiene.*	7.2
Figure 2	15	*Promoting good hand hygiene practice is as important as providing good hand hygiene resources*	6.6
Figure 3	16	*Employees should be given clear guidance on good practice for hand hygiene*	5.2
Figure 2	17	*Money spent on hand hygiene facilities is a good investment*	4.6
Figure 3	18	*Good hand hygiene is a personal matter, so employers should not interfere.*	3.3

## Data Availability

The data presented in this study are available on request from the corresponding author. The data are not yet publicly available.

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
