# Peer review of "Smart Hand Sanitisers in the Workplace: A Survey of Attitudes towards an Internet of Things Technology"

_ijerph, 2022, doi:10.3390/ijerph19159531_

Round 1

Reviewer 1 Report

The paper is well written and provides good analysis of participant feedback on this survey into user feedback to smart hand sanitizer acceptance.

The paper uses both the terminology COVID-19 and Covid-19 - I don't know which is correct but it would be good to have consistency. 

I think there is potentially a large variance in what a smart hand sanitizer might do and how it might operate. It might be good to articulate these further (e.g. using cameras with facial recognition, sensors to detect empty hand sanitizer, timestamping each time sanitizer is dispensed).

I also wonder how effective the facial recognition will be if people are wearing masks as many have done so during the pandemic.

I wonder if a smart sanitiser could help differentiate between frequency (of sanitizing) and quality of the sanitizing (how well it was used vs how frequently it was used)

My understanding is that hand sanitizer performs much less well (than soap and warm water) in preventing gastro, so I wonder how this could be addressed in a wider monitoring system to increase the value of such a system.

I also wonder if there is a more effective means of encouraging better hand hygiene than using smart sanitizers (possibly reminder audio or video reminder messages each time someone is detected as being in the toilet).

Author Response

We would like to thank reviewer 1 for their encouraging comments about our article, and for their interesting observations.  In the light of Reviewer 1’s observations, we have commented on the role of hand gel as a hand hygiene resource in addition to soap and water, we have further explored some of the applications of Smart sanitisers, and have edited part of the discussion. We have also standardised terminology when referring to COVID.

The most significant revisions to the earlier version of the paper have been highlighted in yellow.

Reviewer 2 Report

Please find the comments and suggestions in the attached file.

Author Response

We would like to thank reviewer2 for taking the time to read our article, and for their detailed and helpful comments.  The paper has been substantially revised in the light of these suggestions.   The most significant amendments have been highlighted in yellow.  Section 1 has now been substantially edited to address points 1 to 3.  The research gap has been more clearly described, and the novelty of the article has been emphasised.  The part of Section 1 dealing with non-pharmaceutical interventions has been rewritten to highlight its relevance to Smart sanitisers.  

We were encouraged by Point 4 (b): “I think your research has a wider context… than just smart sanitisers and the application of IoT technologies.”  However, our aim in formulating the survey was to explore environmental factors that might impact on the acceptability of Smart sanitisers in the workplace.   As the reviewer noted, the survey (Appendix 1) comprised 21 items.  Some of these were simple questions (eg, Item 1: How old are you?). Many though, were multiple sets of Likert-style statements.  Items 18 and 19 elicited responses to 10 Likert-style statements which dealt explicitly with attitudes to data monitoring by Smart sanitisers.  Responses to the other 28 Likert-style statements provided us with insights into opinions and circumstances which may influence workers’ attitudes to Smart sanitisers.  Hence, although we are gratified by the reviewer’s opinion that our findings have wider relevance than is indicated by the title of the paper, the current title does accurately reflect the purpose and the focus of our research.

All the observations made in point 5 have been addressed in the revised text.  Some of the older references refer to items of relevant historical or methodological relevance, but where possible, recent articles have been cited, with the result that 50% of the articles now referenced were published in the last 5 years and most were published in the last 10.

Points 5 to 10 usefully drew attention to omitted details.  The omitted information has now been provided.

Errors identified in points 11 to 13 have been corrected.

Round 2

Reviewer 2 Report

Dear Authors,

Most of the points were addressed in the revised version of the manuscript. Therefore, I think the article improved significantly and can be published as is.

Good luck in future research!